# Plasma Adipokines Profile in Prepubertal Children with a History of Prematurity or Extrauterine Growth Restriction

**DOI:** 10.3390/nu12041201

**Published:** 2020-04-24

**Authors:** María Dolores Ordóñez-Díaz, Mercedes Gil-Campos, Katherine Flores-Rojas, María Carmen Muñoz-Villanueva, Concepción María Aguilera-García, María Jose de la Torre-Aguilar, Juan Luis Pérez-Navero

**Affiliations:** 1Unit of Neonatology, Department of Paediatrics, Maimonides Institute for Biomedical Research (IMIBIC), Reina Sofia Hospital, University of Córdoba, 14005 Córdoba, Spain; mdordonezdiaz@gmail.com; 2Unit of Metabolism and Paediatric Research, Maimonides Institute for Biomedical Research (IMIBIC), Reina Sofia University Hospital, University of Córdoba, 14005 Córdoba, Spain; katherine1.flores@gmail.com (K.F.-R.); delatorremj4@gmail.com (M.J.d.l.T.-A.); juanpereznavero@hotmail.com (J.L.P.-N.); 3Biomedical Research Center—Pathophysiology of Obesity and Nutrition (CIBEROBN), Carlos III Health Institute, 28029 Madrid, Spain; caguiler@urg.es; 4Unit of Methodology in Investigation, Maimonides Institute for Biomedical Research (IMIBIC), 14005 Córdoba, Spain; villanuevamcm@gmail.com; 5Department of Biochemistry and Molecular Biology II, Institute of Nutrition and Food Technology, Center of Biomedical Research, Lab 123, University of Granada, 18071 Granada, Spain; 6Biomedical Research Center—Rare Diseases (CIBERER), Carlos III Health Institute, 28029 Madrid, Spain

**Keywords:** metabolic syndrome, programming, adiponectin, leptin, resistin, prematurity, extrauterine growth restriction

## Abstract

Adipose tissue programming could be developed in very preterm infants with extrauterine growth restriction (EUGR), with an adverse impact on long-term metabolic status, as was studied in intrauterine growth restriction patterns. The aim of this cohort study was to evaluate the difference in levels of plasma adipokines in children with a history of EUGR. A total of 211 school age prepubertal children were examined: 38 with a history of prematurity and EUGR (EUGR), 50 with a history of prematurity with adequate growth (PREM), and 123 healthy children born at term. Anthropometric parameters, blood pressure, metabolic markers and adipokines (adiponectin, resistin, leptin) were measured. Children with a history of EUGR showed lower values of adiponectin (μg/mL) compared with the other two groups: (EUGR: 10.6 vs. PREM: 17.7, *p* < 0.001; vs. CONTROL: 25.7, *p* = 0.004) and higher levels of resistin (ng/mL) (EUGR: 19.2 vs. PREM: 16.3, *p* =0.007; vs. CONTROL: 7.1, *p* < 0.001. The PREM group showed the highest values of leptin (ng/mL), compared with the others: PREM: 4.9 vs. EUGR: 2.1, *p* = 0.048; vs. CONTROL: 3.2, *p* = 0.029). In conclusion, EUGR in premature children could lead to a distinctive adipokines profile, likely associated with an early programming of the adipose tissue, and likely to increase the risk of adverse health outcomes later in life.

## 1. Introduction

Very low birth weight (VLBW, <1500 g and/or gestational age (GA) <32 weeks) neonates comprise approximately 5% of all of births in European countries [1]. Despite improvements in the quality of perinatal care, prematurity continues to be a serious health problem due to their increased risk of sequelae and long-term disabilities [2], including metabolic complications, such as insulin resistance (IR), type 2 diabetes (T2D), metabolic syndrome (MS), or altered cardiovascular functions [3,4]. Early life “programming” as an adaptive response to insults during the perinatal time might lead to a variable persistent impact at later ages [5,6]. However, the pathophysiological mechanisms underlying this programming are yet to be clarified. Nutritional exposure and postnatal growth might be influential factors to consider [6].

Preterm infants, especially those with lower GA and birth weight are markedly at risk of extrauterine growth restriction (EUGR) and may develop postnatal weights below the 10th (P10) or 3rd percentile (P3) at discharge or at 36 weeks of postmenstrual age (PMA) [7,8,9]. The EUGR condition is associated not only with later unfavorable growth rates and neurodevelopmental impairment, but also with alterations in cardiometabolic risk markers, including selected adipokines [10], and other long-term health complications [11]. Reviews focused on the impact of adipose tissue disturbance in the adipokine profile have demonstrated that balance shifts to favor proinflammatory adipokines may play a critical role in the development of cardiometabolic pathologies [12,13,14]. It is known that abnormal adipokine profile may be observed in obese children [15]. Moreover, it was suggested that impaired growth during critical periods, such as in the third trimester of pregnancy can lead the programming of adipose tissue, conferring a high subsequent cardiometabolic risk later in life [16]. In fact, an abnormal production of some adipokines was described in children with intrauterine growth restriction (IUGR) [17] who, in turn, may present an elevated risk of inflammatory chronic diseases [4,18]. Similarly, an early postnatal environment with EUGR might be associated with a higher vulnerability to an adipose tissue disturbance [16]. However, there is scarce literature about the role of the antecedents of EUGR in the later adipokines status, although it could be similar to obesity or IUGR condition. In a previous research of this group, children with a history of EUGR exhibited a distinctive adipokine profile with respect to healthy children born at term [19]. However, prematurity condition was not considered to be a possible independent factor involved in the adipokines profile, as some studies reported [20]. It is necessary to clarify whether changes in adipokine levels may be affected by prematurity in itself. Therefore, the aim of this study was to evaluate serum concentrations of selected adipokines (adiponectin, resistin, and leptin) in premature children with EUGR, those without EUGR and healthy children born at term; and to determine the associations between the anthropometric and metabolic parameters and the conditions of EUGR and prematurity at the prepubertal stage.

## 2. Materials and Methods

### 2.1. Study Participants

This is a descriptive, analytical, and observational, cohort study. The sample comprised 211 Caucasians children, all born between 1996 and 2008, evaluated at the prepubertal status (Tanner 1) and school age. Three study groups were established. The first was the EUGR group, with 38 children with the following criteria: prepubertal children born at ≤32 weeks of GA with a weight above P10 for GA at birth, and meeting the definition of “true-EUGR” based on the different concepts proposed in the literature (weight < P3 at 36 weeks-PMA and at discharge from the neonatal unit) [9]. A total of 55 participants were initially recruited. Then, 11 children were excluded due to failure to obtain consent, and 6 other candidates were in puberty after physical examination or collection of hormonal data. There were no dropouts. The PREM group included a total of 50 children without puberty signs born at ≤32 weeks of pregnancy with a weight above P10 for GA at birth, and appropriate postnatal growth defined as weight ≥ P3 at 36 weeks-PMA and at discharge from the neonatal unit. As the EUGR group, PREM children were evaluated at prepubertal age. The control group consisted of 123 prepubertal children meeting the following inclusion criteria: born at term with adequate birth weight for GA (2500–3500 g, 38–42 week’s GA) and born in the same period as children in the other two groups, without underlying conditions and any remarkable disease history. These children attended the hospital’s Outpatients Unit at prepubertal age, on suspicion of mild illness that was later ruled out and requiring blood tests. All the analyses showed that controls were free of any disease. All the children were recruited from a single tertiary hospital, which is commonly community benchmark for healthcare of certain pediatric and neonatal pathologies.

This study was conducted in accordance with the Declaration of Helsinki and was approved by the Institutional Hospital Ethical Committee (protocol no 228, ref. 2466; version 1; 8 April 2014). The selected subjects were incorporated into the study after all the inclusion criteria were fulfilled, and informed written consent from the parents or legal guardians of the children was obtained. Confidentiality of all personal information was protected. Access to medical data was obtained conforming to the hospital ethical standards.

### 2.2. Clinical History and Physical Examination

Perinatal clinical data and personal and family health records were collected and reviewed retrospectively from the clinical history. The mothers of all the children involved were healthy during pregnancy, without a family history of metabolic or cardiovascular diseases. The weight of preterm children was collected at birth, 36 weeks-PMA and at discharge, and the participants in the EUGR and PREM groups were selected according to the weight percentile charts for age, sex and GA developed by Carrascosa et al. [21]. Children with the diagnosis of intrauterine growth restriction (IUGR) or small gestational age (SGA), defined as fetuses or newborns who had failed to achieve normal-weight based on previous growth measurements during the pregnancy, and with an estimated fetal weight that is less than the 10th percentile for GA [22], were excluded in this study. Fetal growth was evaluated using anthropometric centiles for gestational ages 24 to 42 weeks from Yudkin et al. [23].

Preterm participants received similar neonatal cares and the parenteral nutrition protocol established in the neonatology unit. The parenteral nutrition was composed of macro and micronutrients amounts recommended and individualized according to the variables gestational age, weight, days of life, clinical evolution, and laboratory parameters [24]. These preterm also received similar enteral nutrition protocol—initially trophic and then full-feeding with fortified breast milk and/or formula for premature infants. At the time of evaluation, at prepubertal age, these children were free of any acute or chronic disease.

A complete physical examination including weight and height was recorded in all the children using a HEALTH SCALE^®^ ADE RGT-200 stadiometer (ADE, Hamburg Germany), with the subjects barefoot and in minimal clothing. The prepubertal status (Tanner 1) was confirmed with physical exploration and sexual hormone levels (follicle stimulating hormone, luteinizing hormone, estradiol, and testosterone). Body mass index (BMI) and z-score for weight, height and BMI were calculated for all the participants by using the standard growth percentile charts for the Spanish population [25]. A delay in weight or height was defined as weight or height ≤ P10 at the time of evaluation. A delay in weight-height was defined as both weight and height ≤ P10 at the time of evaluation. Obesity was defined as BMI > P97 at the prepubertal stage, according to Cole et al. [26], specific for age and sex and referred to the growth charts by Hernández et al. Systolic (SBP) and diastolic blood pressure (DBP) were measured twice by the same observer with a Dinamap V-100, and their blood pressure (BP) percentiles were established according to the age and sex of the subject [27].

### 2.3. Biochemical and Adipokine Analysis

Blood samples were collected in all the groups of children at 09.00 a.m. after a 12.00-h overnight fast and at rest while lying down and using an indwelling venous line (median cubital vein) to draw a two 3-mL sample, which was used to extract serum and plasma, respectively. Serum samples were analyzed within 2 h of collection, while plasma samples were divided into aliquots and frozen at −80 °C until analysis.

Serum total cholesterol (TC) (CV ≤ 3%), high density lipoprotein cholesterol (HDL-C) (CV ≤ 4%), low density lipoprotein cholesterol (LDL-C), triglycerides (TG) (CV ≤ 5%) and markers of carbohydrate metabolism, glucose (CV ≤ 5%) and insulin (CV ≤ 7%) were measured. IR was calculated by the homeostatic model assessment index (HOMA-IR = insulin (mU/L) × glucose (mmol/L)/22.5). Follicle stimulating hormone, luteinizing hormone, estradiol, and testosterone were analyzed to confirm the prepubertal stage in all the children.

Analyses were performed by standardized laboratory methods and external and internal quality controls were performed according to hospital protocols. Autoanalyzers Architect c16000 and i2000SR (Abbott Diagnostics^®^, Madrid Spain) were used for the measurements of general biomarkers (glucose, TC, HDL-C, LDL-C, TG) and general hormones (insulin), respectively.

To determine adiponectin (CV, 7.9%), resistin (CV, 6.0%), and leptin (CV, 7.9%) (Cat. HADK2–61K-B), LINCOplex kits of human monoclonal antibodies (Linco Research, St Charles, MO, USA) were used on a Luminex 200 System (Luminex^®^ X MAP™ Technology (Labscan™ 100), Luminex Corporation, Austin, TX, USA). This is multi-analyte simultaneous detection system with a cytometer with 96-well filter plates. Each kit used was validated for sensitivity, recovery, linearity, precision, and specificity. An immunoassay on the surface of polyethylene fluorescent-coded microspheres was performed. This surface is covered with a specific capture antibody that selectively binds to the adipokine, simultaneously introducing a marked detection antibody. This reaction mixture is incubated and, via a laser diode signal of each microsphere, is identified and quantified. All the analytes were tested individually and in combination to ensure their compatibility with the multiplex system (Millipore technical protocols). The number of repetitions for the assessments of the general metabolic biomarkers and those of adipokines values were of one and two, respectively.

### 2.4. Dietary Assessment

Standard food frequency intake and a 24-h diet recall method questionnaires were made to collect information on food habits in the three groups at prepubertal age. The computer program “ODIMET^®^ (Organizador Dietético Metabólico)”, designed by the Santiago de Compostela University Clinical Hospital (Spain 2008), was used to estimate the daily energy and fiber intake and dietary macronutrient composition. The dietary information obtained was compared against the charts included in the Guide of Healthy Food Habits designed by the Spanish Society of Community Nutrition [28].

### 2.5. Statistical Analysis

The variables BP, HDL-C, LDL-C, TG, insulin, and HOMA-IR were selected to perform the sample size calculation. Assuming a minimum 30% difference in the mean values of these main variables studied among the three groups, an α-error of 0.05, a β-error of 0.1 in a bilateral contrast of hypotheses, and a loss to follow-up of 15–20%, 37 EUGR children, 37 PREM children and 111 controls (ratio 1:1:3) were need to perform the study. All the results were adjusted for sex, birth weight and age at the prepubertal stage. Moreover, a sub-analysis of the anthropometric parameters, BP levels, metabolic markers and adipokines profile excluding obese patients in all the groups were performed.

Variables are reported as proportion, mean values and standard deviations, or median values and interquartile ranges. The Shapiro-Wilk test was used to evaluate normal data distribution, and homogeneity of variance was estimated by the Levene’s test. Proportions were compared with the χ^2^ test. Comparison of quantitative variables among the three groups was performed by the ANOVA or Kruskal-Wallis tests, and between the two groups by the Student’s *t* test or the Mann–Whitney U test. Spearman’s rank correlation coefficients were calculated to assess the relationship between all the variables recruited. The relations between EUGR and prematurity conditions with anthropometric parameters, BP, metabolic biomarkers and adipokines changes at prepubertal age were explored using a logistic regression model, estimating odds ratios (OR) values and 95% confidence intervals (95% CI). The variables that showed an association with a value of *p*< 0.25 were used for the multiple logistic regression analysis. By the method of backward method selection, the variables with values of *p* ≥ 0.15 for the Wald statistic were eliminated one by one from the model until obtaining the estimate of the adjusted OR.Data analysis was carried out using the software PASW Statistics 18 (IBM SPSS, Inc, IBM, New York. USA. *p* was significant at <0.05.

## 3. Results

The most relevant antecedents of the perinatal stage of preterm children with EUGR and with adequate growth are summarized in Table 1. Although EUGR children presented lower birth weight than PREM children, both groups had birth weight percentiles above P10 for GA according to the inclusion criteria. Demographic and anthropometric parameters, BP, and general biochemical markers in all children at the prepubertal age are shown in Table 2.

Regarding nutritional assessment, the Dietary Reference Intakes (DRIs) and the comparison by groups in macronutrients intake are shown in Table 3. Daily caloric intake was not different between groups. The PREM group showed the highest carbohydrate intake (*p* < 0.001), followed by the EUGR group (*p* = 0.010) and then the Control group. Lipid intake was higher in the EUGR group than in the PREM group (*p* < 0.001), and similar to the CONTROL group (*p* = 0.785). Protein intake in the EUGR group was similar to that in the PREM group (*p* = 0.208), and higher than in the Control group (*p* < 0.001).

Plasma adipokine levels displayed differences among the three groups (Figure 1A–C). Prepubertal EUGR children exhibited the lowest levels of adiponectin and highest levels of resistin, compared with the other two groups. Children in the PREM group also showed lower values of adiponectin and higher values of resistin than controls, and these differences maintained after the sub-analysis excluding the five obese children from PREM group (adiponectin in PREM without obese children: 18.3 μg/mL vs Control: 25.7 μg/mL, *p* < 0.005; levels of resistin in PREM without obese children: 15.6 ng/mL vs. Control: 7.1 ng/mL, *p* < 0.001). In plasma leptin, the PREM group showed the highest values, compared with the other groups, without differences in the EUGR and control groups. These differences in leptin values disappeared when the obese children from PREM group were excluded (PREM without obese children: 3.8 ng/mL vs. EUGR: 2.1 ng/mL vs. CONTROL: 3.2 ng/mL, *p* = 0.192).

After the correlation analysis in the whole study sample, correlations among plasma adipokines and anthropometric measurements, BP or metabolic parameters in different subgroups were made. Correlations that were significant are indicated in Table 4. Adiponectin showed a positive correlation with HDL-C levels in the PREM group and, especially, in the EUGR group. Resistin values presented a positive correlation with LDL-C values only in the EUGR group. Although positive associations between leptin and BMI z-score and waist circumference were observed in the three groups, the strongest correlations were found in the EUGR group (Table 4). Similarly, leptin positive associations with values of TG, insulin and HOMA-IR were stronger for prepubertal children with the EUGR condition than for the other children (Table 4). Concentrations of leptin were positively correlated with BP levels in the PREM subjects, but not in the other two groups, and negatively correlated with HDL-C in the PREM and control groups (Table 4).

Table 5 displays the logistic regression analyses performed to identify the association between the anthropometry, BP, and metabolic variables (including adipokines) of prepubertal children with the antecedents of prematurity and the EUGR condition. The multivariable analysis showed that a decrease in the BMI z-scores and adiponectin concentrations at prepubertal age were statistically correlated with prematurity only when there was a history of EUGR. Leptin levels were positively associated with prematurity only when there was an adequate postnatal growth. Both conditions, prematurity and EUGR, were linked to decreased HDL-C and increased SBP and resistin values at prepubertal age.

## 4. Discussion

The current results provide new differential information about the influence of prematurity and early postnatal growth restriction on metabolic status and the adipokines status during childhood, and suggest a possible early programming in the AT of premature children. Our study demonstrates that children with a history of prematurity show higher leptin and resistin concentrations and lower adiponectin levels than healthy children. Most of these differences are more significant in children affected by an early EUGR. Furthermore, these adipokines were correlated with parameters such as BP in premature children, and with several markers of lipid and hydrocarbon metabolism particularly in children with a history of EUGR. These results could lead an increased risk for metabolic and cardiovascular diseases in adult life.

Adipokines are bioactive molecules similar to cytokines in structure, mainly secreted by pre- and mature cells derived from AT in response to changes in adipocyte glycerol storage and inflammation [29]. These biological mediators have pro- and anti-inflammatory properties and modulate several physiological functions, such as energy balance, insulin sensibility, inflammatory and immune responses, appetite, and vascular homeostasis [30]. Balance between adipokines is crucial to integrate metabolism with several physiologic functions [13], and proinflammatory adipokines may play a critical role in the progression of chronic inflammation and the development of MS, hypertension, T2D, obesity and CVD [12,13,31,32,33], even at early ages [15,34].

Many different factors may influence adipokine values, such as race, BMI, and the degree of adiposity [35]. Although dietary consumption during childhood may be determinant in the growth and adult health [36], in the present study, diet or BMI seems unrelated with the results in plasma adipokines, at least in EUGR group, in which there was no obese and there were not relevant nutritional results since there were no differences in energy intake. EUGR children exhibited more changes in adipokines levels but maintained lower anthropometric values, and had lower BMI z-scores than those without EUGR or control children. Therefore, the changes in adipokines levels in premature children may be induced by an adipose tissue dysfunction probably related to the poor growth during the neonatal period, rather than bytheir dietary intake or values of BMI. This premise is consistent with previous studies in non-obese children indicating there were no associations between dietary factors and adipokines [37]. Moreover, a recent study evaluating whether the dietary pattern differs between healthy normal-weight children and normal-weight children suffering from adipokines changes found no differences regarding the intake of total calories and macronutrients between the two groups [38]. In contrast, a higher proportion of obesity was found in premature children without EUGR than in the other children. It may indicate that the role of prematurity in adipose tissue programming and the pathways to develop metabolic alterations isdifferent in these groups. A rapid catch-up growth later in preterm children without EUGR, who exhibited a similar BMI z-score that those healthy born at term, could have led to an increased adiposity and an abnormal body composition, resulting in changes in adipokines values. In fact, associations between a rapid childhood weight gain and higher adiposity and risk metabolic markers or alterations in adipokines values were reported in preterm children and adolescents [20,39].

Sex and age may be influencing factors in adipokine levels [40], and some recent research also highlighted that normal physiological changes of puberty itself may have an impact on levels of some adipokines [41]. EUGR children were predominantly male and showed a lower absolute birth weight (without reaching the 10th percentile) than PREM children. These data may be expected, since a lower birth weight and male sex have been recently independently related to the presence of EUGR in [9]. We selected school aged children to avoid this puberty influence. In our study, the EUGR and control groups were older than the PREM group, but all the participants were at school age and without pubertal signs, according to the inclusion criteria. Moreover, allour results were adjusted for sex and age at the time of evaluation in prepubertal age.The differences in prenatal steroid use and mechanical ventilation between EUGR and PREM children could result in changes in adipokines levels especially during the perinatal period and maybe at the prepubertal stage. For this reason, we considered these variables and to follow if there were any sequelae in these children related. Notwithstanding, it was reported that values of adiponectin, leptin or resistin are not been affected by steroid treatment [42] or mechanical ventilation [43] in critical ill adult patients.

Adipokines values may also differ by perinatal factors, such as weight and GA at birth [44,45]. Lower values of adiponectin, leptin and resistin were reported in infants born preterm [44,46], and in those small for gestational age [44,45], compared with full-term infants and newborns adequate for gestational age (AGA). However, adipokine research beyond birth in premature children and its link to early postnatal growth is scarce and controversial. Some comparative studies reported that adiponectin levels remained altered after term-equivalent age [47,48], and at discharge [49] in preterm infants. Nakano et al. [47] studied adiponectin changes in preterm children and found a significant decrease in values over time, with lower values at term-and at 6 month-equivalent ages.

Significantly lower adiponectin concentrations in SGA children, even with catch-up growth, compared with AGA children, were also noted [50]. A fat cell hypertrophy, paradoxically associated with decreased adiponectin expression in adipocytes, was implicated as a mechanism [33]. On the other hand, some reports noted that adiponectin levels elevate substantially with improved growth at postnatal age of premature infants [51]. Similarly, findings of Flexeder et al. [52] showed a positive association between the peak weight velocity from birth to 2 years old and adiponectin levels 10 years later among boys.

A decrease in adiponectin values may be expected in premature children with early postnatal growth restriction compared with those that grow accordingly. In accordance with this premise, some authors reported lower adiponectin concentrations in premature children with EUGR, compared with healthy children, but without discerning the influence of both prematurity and the EUGR condition on results [19]. In the present study, the EUGR group which exhibited lower z-scores for BMI than PREM children also showed lower values of adiponectin. Moreover, in the adjusted regression analysis, adiponectin showed a consistent inverse association with prematurity condition only when EUGR was associated. These results could indicate that in preterm children, the aptitude of adipocytes and adiponectin secretion might be affected by an early stunting.

Like adiponectin, lower leptin levels were detected in premature infants at birth, with greater decreases in relation to lower GA [46]. However, this correlation may become inverted when levels are analyzed at later ages, probably influenced by the growth pattern and an increase in BMI [53] and the amount of fat mass [54]. In line with these statements, some studies demonstrated that a rapid weight catch-up in early infancy, often detected in premature and SGA children, may affect both, the degree of adiposity and the IR, as well as leptin concentrations [55]. Toprak et al. [56] and Kawamata et al. [57] reported an increase in leptin levels in preterm infants after 30 days of life and aftersix weeks of follow-up, respectively, accompanied by an increase in body weight and subcutaneous AT [56]. However, in another study [58], leptin levels did not differ significantly among children born preterm or full-term, and no association between leptin and catch-up growth was observed. In our study, the PREM group exhibited increased leptin levels as well as higher values of insulin and HOMA-IR, compared with the control group. In the present study, the PREM group exhibited increased leptin levels as well as higher values of insulin and HOMA-IR, compared with the control group. However, these differences were not found when the five obese children of the PREM group were excluded from the analysis, suggesting a higher risk of fat tissue remodeling probably related with increased BMI z-score in this group of children. In fact, the prevalence of the obesity has been associated with prematurity, and it was reported that preterm children with higher adiposity and metabolic abnormalities, may have disproportionately higher values of fat mass and leptin [4,59].

Up to the present, the available literature focusing on the relationship between prematurity and resistin levels has been limited and inconsistent. In a study by Martos-Moreno et al. [48], preterm neonates had higher resistin levels than full-term neonates. Conversely, some authors observed decreased resistin concentrations in preterm infants, compared with full-term infants [60], whereas resistin levels were not correlated with GA in another study [61]. Beyond birth, some reports found that resistin levels were similar between preterm and full-term children at school age [53,62]. However, in our study, preterm prepubertal children exhibited higher resistin levels than term children, and resistin values were positively associated with prematurity. Regarding the role of postnatal growth in resistin levels, the results obtained in studies have also been contradictory. Some authors affirmed that SGA infants presented lower resistin concentrations than AGA infants [63], whereas other authors observed higher values [64]. Korhonen et al. [62] did not find an association between resistin concentrations and growth among 6- to 14-year-old former VLBW children. In our study, resistin concentrations were higher in EUGR preterm children than in healthy children, as reported earlier [19], but also higher than in PREM children. Therefore, our findings demonstrated that both preterm birth and EUGR might have an effect on AT dysfunction and higher resistin levels in childhood.

Previous studies showed the role of adipokines in metabolic and cardiovascular diseases [13,34]. In obese children and adolescents, hypoadiponectinemia and hyperleptinemia, as well as higher values of resistin were well correlated with an atherogenic lipid profile and markers of IR [65,66]. The majority of these correlations have also been documented in term SGA children [50,67]. However, the literature on the role of adipokines in regulating lipid metabolism and/or insulin sensibility in preterm children is limited [20]. Nagasaki et al. [68] and Kistner et al. [69] reported positive correlations between leptin and insulin concentrations in AGA preterm infants and premature children at the start of puberty, respectively. The authors suggested impaired fat mass accumulation in these subjects but have not evaluated the role of early postnatal growth in findings. To our knowledge, no other research has been carried out showing relationships between adipokine values and cardiometabolic markers in prepubertal EUGR children. Children with a history of prematurity who were not obese showed lower HDL-C values than controls, and decreased HDL-C levels were correlated with lower adiponectin concentrations, especially in children with EUGR. The EUGR group also showed the strongest positive associations between leptin and resistin levels, and markers of lipid and hydrocarbon metabolism.

It must be also highlighted that a significant inverse correlation between adiponectin and resistin in EUGR children was found. The interdependency between plasmatic adipokines’s profiles is not a common subject of study [70]. It was reported that the relation adiponectin/resistin index, may be a useful integrated determinant for diagnostic of insulin resistance or metabolic syndrome [71] and appears to play a role in inflammation pathways [13]. In fact, adiponectin exerts an anti-inflammatory protective effect, whereas resistin may have proinflammatory properties [72]. Thus, an inverse correlation between both adipokines leading to hypoadiponectinemia and hyperesistinemia may be expected in aproinflammatory status as could occur in prematurity and impaired growth.

A limitation to this study concerns the lack of the evaluation of body fat mass, which may be an influential factor of the ranges of adipokines [35]. Some authors reported that preterm infants have a greater percentage of adiposity at term-equivalent age, compared with term infants [73]. Therefore, this disorder might influence later metabolic outcomes. However, the current study presents several strengths. First, it benefits from a large group of children from a same hospital, a careful selection of the children, as well as from the exhaustiveness to diagnose EUGR. In addition, we exclusively selected school aged children to avoid puberty influence in all the metabolic and anthropometric parameters that can be affected by hormone status. Second, prior studies show ambiguous results regarding the mechanisms involved in the influence of prematurity on long-term health, but in this study, comparison of data was performed from three well-defined groups of children, which may help to assess such mechanisms. Third, unlike other studies, we analyzed both perinatal factors and long-term data of preterm children. It includes comprehensive standardized measurements of biochemical variables and, more importantly, this is a pioneer study of adipokines conducted with a group of premature children, with and without EUGR, compared with healthy children.

## 5. Conclusions

In conclusion, EUGR in premature children exhibited a distinctive adipokine profile, probably associated with an early programming of the adipose tissue, which could increase the risk of adverse health outcomes later in life. Based on our results and considering the high prevalence of EUGR, more studies detailing the profile of the main adipokines in premature with a postnatal under-growth will permit a better knowledge of their adipose tissue status and the pathogenesis of the possible EUGR-linked morbidities in preterm children.

## Figures and Tables

**Figure 1 nutrients-12-01201-f001:**
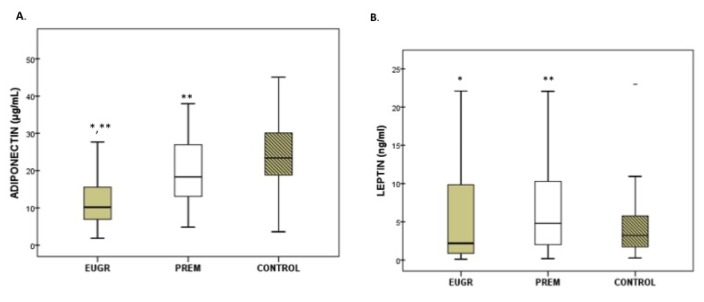
Concentration of adiponectin (**A**), leptin (**B**) and resistin (**C**) in prepubertal children with a history of prematurity and extrauterine growth restriction (EUGR) (*n* = 38), premature without EUGR (PREM) (*n =* 50) and control children (*n =* 123). Values are expressed as medians and interquartile ranges. * Median value was significantly different from that of the PREM group with *p* < 0.05. ** Median value was significantly different from that of the CONTROL group with *p* < 0.05 (Mann–Whitney *U* test). *p*, probability adjusted for gestational age, birth weight and length. Adiponectin: EUGR vs. PREM *p* < 0.001, EUGR vs. CONTROL *p* < 0.001; PREM vs. CONTROL *p* = 0.004. Leptin: EUGR vs. PREM *p* = 0.048, EUGR vs. CONTROL *p* = 0.214, PREM vs. CONTROL *p* = 0.029. Resistin: EUGR vs. PREM *p*= 0.007, EUGR vs. CONTROL *p* < 0.001, PREM vs. CONTROL *p* < 0.001.

**Table 1 nutrients-12-01201-t001:** Perinatal data of children with a history of prematurity and extrauterine growth restriction (EUGR group) and those with prematurity without EUGR (PREM group).

Perinatal Data	EUGR Group (*n* 38)	PREM Group (*n* 50)	*p*-Value
Gestational age (weeks)	29.5 (25.0, 32.0)	29.0 (25.0, 32.0)	0.645 *
Birth weight (g)	1100.0 (660.0, 1707.0)	1290 (796,1510)	0.041 *
Multiple pregnancy (%)	31.6	42.9	0.543 ^‡^
Prenatal corticosteroids (%)	81.3	57.9	0.003 ^‡^
Cesarean delivery (%)	65.8	56.0	0.619 ^‡^
Apgar test score at minute 1	5.3 ± 2.8	6.5 ± 1.7	0.101 ^†^
Apgar test score at minute 5	7.6 ± 2.1	7.7 ± 2.3	0.988 ^†^
Hyaline membrane disease (%)	42.1	36.0	0.707 ^‡^
Mechanical ventilation (%)	16.8	64	0.001 ^‡^
Patent ductus arteriosus (%)	23.7	12.0	0.147 ^‡^
Necrotizing enterocolitis (%)	7.9	6.0	0.999 ^‡^
Bronchopulmonary displasia (%)	23.7	13.0	0.205 ^‡^
Cerebral hemorrhage (%)	21.1	12.2	0.52 ^‡^
Weight at 36 weeks-postmenstural age (g)	1769.4 ± 149.6	2181.6 ± 213.7	<0.001 ^†^
Weight at discharge (g)	2475.0 (2245.0, 3200.0)	2455.0 (2230.0, 3895.0)	0.923 *

Data are expressed as percentages of subjects (%), mean values (±standard deviations) or medians (interquartile ranges). Statistical significance obtained by * Mann–Whitney *U* test, ^‡^ χ^2^ test and ^†^ Student’s *t* test.

**Table 2 nutrients-12-01201-t002:** Demographic and anthropometric parameters, blood pressure and biochemical markers in prepubertal children with a history of prematurity and extrauterine growth restriction (EUGR group), those with prematurity without EUGR (PREM group) and healthy children (Control group).

Prepubertal Data	EUGR Group (*n* 38)	PREM Group (*n* 50)	Control Group (*n* 123)	*p*-Value
Age (years)	9.0 ^a^ (3.0, 13.0)	7.5 ^b^ (4.0, 12.0)	9.0 ^a^ (6.0, 12.0)	<0.001 *
Sex (M/F) (%)	71.1/28.9 ^a^	52.0/48.0 ^b^	47.9/52.1 ^b^	<0.05 ^‡^
BMI z-score §	−0.6 ^a^ (−2.3, 1.7)	−0.4 ^ab^ (−2.0, 3.5)	−0·2 ^b^ (−1.2, 0.8)	0.012 *
WC (cm)	57.5 ^a^ (43.5, 83.0)	59.0 ^a^ (46.0, 88.0)	58.0 ^a^ (22.5, 90.0)	0.367 *
Obesity (%)	0.0 ^a^	10.0 ^b^	0.0 ^a^	<0.001 ^‡^
Delay weight-height (%)	13.2 ^a^	4.0 ^b^	0.0 ^c^	<0.001 ^‡^
Delay weight (%)	21.0 ^a^	6.0 ^b^	0.0 ^c^	<0.001 ^‡^
Delay height (%)	24.0 ^a^	4.0 ^b^	0.0 ^c^	<0.001 ^‡^
SBP (mmHg)	114.0 ^a^ (86.0, 138.0)	101.5 ^b^ (62.0, 129.0)	90.0 ^c^ (48.0, 119.0)	<0.001 *
DBP (mmHg)	72.5 ^a^ (38.0, 89.0)	58.0 ^b^ (34.0, 75.0)	59.0 ^b^ (35.0, 84.0)	<0.001 *
SH (%) ||	46.0 ^a^	10.0 ^b^	3.0 ^c^	<0.001 ^‡^
DH (%) ||	37.0 ^a^	0.0 ^b^	3.0 ^c^	<0.001 ^‡^
HDLc (mmol/L)	1.5 ^a^ ± 0.3	1.4 ^a^ ± 0.3	1.7 ^b^ ± 0.3	<0.001 ^†^
LDLc (mmol/L)	2.4 ^ab^ ± 0.5	2.7 ^a^ ± 0.5	2. 4 ^b^ ± 0.6	0.01 ^†^
TC (mmol/L)	4.2 ^a^ ± 0.6	4.4 ^a^ ± 0.6	4.4 ^a^ ± 0.7	0.314 ^†^
TG (mmol/L)	0.6 ^a^ (0.3, 1.6)	0.6 ^a^ (0.4, 2.3)	0.6 ^a^ (0.2, 1.2)	0.514 *
Glucose (mmol/L)	4.9 ^a^ (3.9, 6.3)	4.7 ^b^ (3.8, 5.7)	4.6 ^b^ (3.6, 5.8)	<0.05 *
Insulin (pmol/L)	32.9 ^a^ (11.5, 113.2)	47.3 ^b^ (18.6, 187.7)	35.8 ^a^ (10.7–150.4)	<0.05 *
HOMA-IR ¶	1.0 ^ab^ (0.3, 3.6)	1.3 ^a^ (0.5, 5.6)	1.0 ^b^ (0.3–5)	0.011 *

BMI, body mass index; WC, waist circumference; SBP, systolic blood pressure; DBP, diastolic blood pressure; SH, systolic hypertension; DH, diastolic hypertension; HDLc, high density lipoprotein cholesterol; LDLc, low density lipoprotein cholesterol; TC, total cholesterol; TG, triglycerides; HOMA-IR, Homeostatic Model Assessment Index. Data are expressed as percentages of subjects (%), mean values (±standard deviations) or medians (interquartile range). ^a,b,c^ Values within a row with unlike superscript letters were significantly different (*p* < 0.05), with the *p*-values expressed. Statistical significance obtained by * Kruskal-Wallis and Mann–Whitney *U* tests, ^†^ ANOVA and Student’s *t* tests, ^‡^ χ^2^ test. § BMI was calculated as weight in kg divided by height in m2. || A delay in weight, height or weight-height were defined as weight, height or both weight-height ≤ P10 for the age and sex, respectively. Obesity was defined as BMI > P97 for age and sex. SH and DH were defined as values of SBP and DBP ≥ percentile 95 for the age and sex of the subjects, respectively. HOMA-IR was calculated as insulin in mU/Lmultiplied by glucose in mmol/L and divided by 22.5.

**Table 3 nutrients-12-01201-t003:** Energy and macronutrients intake (% of energy) in prepubertal children with a history of extrauterine growth restriction (EUGR group), those without EUGR (PREM group) and healthy children (control group).

Dietary Intake	DRIs	EUGR Group (*n* 38)	PREM Group (*n* 50)	Control Group (*n* 123)	*p*-Value
Energy (Kcal/day)	2000 *	1855.8 ± 546.2 ^a^	1755.5 ± 510.5 ^a^	1585.5 ± 122.5 ^a^	0.546 ^†^
Carbohydrates (%)	45–60	46.8 ± 5.4 ^a^	58.5 ± 4.8 ^b^	38.7 ± 2.9 ^c^	<0.011 ^‡^
Lipids (%)	20–35	34.9 ± 6.2 ^a^	22.3 ± 5.4 ^b^	33.9 ± 3.3 ^a^	<0.001 ^‡^
Proteins (%)	NS	18.3 ± 3.2 ^a^	19.3 ± 3.5 ^a^	13.5 ± 2.3 ^b^	<0.001 ^‡^

DRIs: dietary reference intakes; NS: not specified. http://www.efsa.europa.eu/; * Energy intake calculated for a moderated physical activity for a range age between 6–9 years; ^a,b,c^ Values within a row with unlike superscript letters were significantly different (*p* < 0.05), with the *p*-values expressed. Statistical significance obtained by ^†^ ANOVA and Student’s *t* tests, ^‡^ χ^2^ test.

**Table 4 nutrients-12-01201-t004:** Results of the correlation analysis among adipokines and anthropometry, blood pressure, and biochemical parameters in the global sample and those in children with a history of prematurity and extrauterine growth restriction (EUGR group), those with prematurity without EUGR (PREM group) and healthy prepubertal children (control group).

Adipokines	Parameters	Global Sample (*n* 211)		EUGR Group (*n* 38)		PREM Group (*n* 50)		Control Group (*n* 123)	
		CC	*p*-Value	CC	*p*-Value	CC	*p*-Value	CC	*p*-Value
**Adiponectin**	SBP	−0.387	**<0.001**	0.270	0.106	−0.214	0.135	−0.114	0.220
	DBP	−0.283	**<0.001**	−0.045	0.788	−0.017	0.909	−0.100	0.280
	HDL-C	0.248	**<0.001**	0.331	**0.042**	0.269	**0.037**	−0.042	0.649
	Glucose	−0.168	**0.015**	−0.082	0.626	0.080	0.579	−0.100	0.272
	Resistin	−0.312	**<0.001**	−0.348	**0.032**	−0.079	0.586	0.279	**0.002**
**Resistin**	SBP	0.441	**<0.001**	−0.109	0.519	0.106	0.466	0.060	0.521
	DBP	0.241	**<0.001**	0.109	0.516	0.003	0.985	−0.035	0.710
	HDL-C	−0.400	**<0.001**	−0.253	0.126	−0.073	0.613	−0.168	0.064
	LDL-C	0.155	**0.025**	0.355	**0.029**	−0.054	0.708	0.114	0.212
	Glucose	0.155	**0.025**	0.012	0.944	−0.059	0.686	−0.014	0.881
**Leptin**	BMI z-score	0.616	**<0.001**	0.829	**<0.001**	0.745	**<0.001**	0.458	**<0.001**
	WC	0.651	**<0.001**	0.796	**<0.001**	0.792	**<0.001**	0.515	**<0.001**
	SBP	0.262	**<0.001**	0.203	0.228	0.410	**0.003**	0.078	0.400
	DBP	0.147	**0.035**	0.106	0.526	0.455	**0.001**	0.042	0.650
	TG	0.393	**<0.001**	0.630	**<0.001**	0.420	**0.002**	0.222	**0.014**
	TC	−0.040	**0.560**	0.321	**0.049**	−0.035	0.810	0.016	0.858
	HDL-C	−0.277	**<0.001**	0.038	0.821	−0.538	**<0.001**	−0.274	**0.002**
	Glucose	0.163	**0.018**	0.165	0.323	−0.067	0.641	0.304	**<0.001**
	Insulin	0.593	**<0.001**	0.692	**<0.001**	0.656	**<0.001**	0.450	**<0.001**
	HOMA-IR	0.572	**<0.001**	0.690	**<0.001**	0.612	**<0.001**	0.454	**<0.001**
	Resistin	0.160	**0.02**	0.176	0.291	−0.027	0.851	0.128	0.161

HDL-C, high density lipoprotein cholesterol; LDL-C, low density lipoprotein cholesterol; BMI, body mass index; WC, waist circumference; SBP, systolic blood pressure; DBP, diastolic blood pressure; TG, triglycerides; TC, total cholesterol; HOMA-IR, homeostatic model assessment-insulin resistance index. Data are expressed as Spearman rho CC and *p*-value. Statistically significant correlations with *p*-value < 0.05 are shown in bold font.

**Table 5 nutrients-12-01201-t005:** Results of multiple logistic regression analysis to identify variables in prepubertal children associated with the condition of prematurity without extrauterine growth restriction (PREM group) and prematurity with extrauterine growth restriction (EUGR group).

	EUGR Group (*n* 38)		PREM Group (*n* 50)	
Parameters	Adjusted OR (95% CI)	*p*-Value	Adjusted OR (95% CI)	*p*-Value
Z-score BMI	0.304 (0.131, 0.705)	**0.006**	0.658 (0.371, 1.167)	0.152
SBP	1.198 (1.121, 1.279)	**<0.001**	1.075 (1.033, 1.119)	**<0.001**
HDL-C	0.919 (0.874, 0.967)	**0.001**	0.916 (0.882, 0.950)	**<0.001**
Adiponectin	0.824 (0.682, 0.997)	**0.046**	1 (0.980, 1.010)	0.963
Leptin	1.148 (0.847, 1.557)	0.373	1.337 (1.128, 1.548)	**0.001**
Resistin	1.546 (1.186, 2.015)	**0.001**	1.59 (1.309, 1.932)	**0.001**

BMI, body mass index; SBP, systolic blood pressure; DBP, diastolic blood pressure; HDL-C, high density lipoprotein cholesterol. Data are expressed as adjusted odd ratio (OR) values and 95% confidence intervals (CI). Statistically significant *p*-value < 0.05 are shown in bold font.

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
