# Peer review of "Plasma Adipokines Profile in Prepubertal Children with a History of Prematurity or Extrauterine Growth Restriction"

_nutrients, 2020, doi:10.3390/nu12041201_

Round 1
Reviewer 1 Report
Adipose tissue programming could be developed in very preterm infants with extrauterine growth restriction (EUGR), with an adverse impact on long-term metabolic status. The cohort study was designed to evaluate the difference in levels of plasma adipokines in children with and without a history of EUGR. A total of 211 children were examined: 38 with a history of prematurity and EUGR (EUGR group), 50 with a history of prematurity with adequate growth (PREM group), and 123 healthy children born at term. Anthropometric parameters, blood pressure, metabolic markers and adipokines (adiponectin, resistin, leptin) were measured at prepubertal age. Children with a history of EUGR showed lower values of adiponectin and higher levels of resistin, compared with the other two groups. The PREM group showed the highest values of leptin. Positive correlations between adiponectin and HDL-C, as well as leptin with the lipid and carbohydrate metabolism parameters were found especially in the EUGR group. A positive association between resistin and LDL-C was observed only in the EUGR group. The authors concluded that EUGR prepubertal children showed a different adipokines profile, likely associated with a perinatal metabolic programming in the adipose tissue.
Major comments:
1. The authors have to clarify whether this study is a “case-control” or “cohort” study. In the Abstract and Methods sections, it is described as case-control study. In the discussion section (P11 Line 315), it became “cohort”.
2. The use of logistic regression analysis in this study is problematic. Logistic regression is a class of regression where the independent variable(s) is (are) used to predict the dependent variable. The dependent variable, therefore, occurred after the independent variable. The time sequence is reversed in the dependent and independent variables in this study. (ie. It is not a case-control study. The EUGR status occurred before the measurements of anthropometric parameters, blood pressure, metabolic markers and adipokines.)
3. In Table 3, correlation analysis should also be performed in the whole study population and then those in the subgroups.
4. In Table 4, multiple logistic regression is an accepted statistical method for assessing the association between an anticedant characteristic (risk factor) and a quantal outcome (probability of disease occurrence), statistically adjusting for potential confounding effects of other covariates. The EUGR status is not an outcome occurred after the risk factors listed in this table.
5. In Table 1 and Table 2, there were differences in birth weight, prenatal steroid use, mechanical ventilation, age at prepubertal measurement and sex distribution between the EUGR and PREM children. The authors have to address whether these differences may result in the changes in prepubertal measurement in their discussion.
6. The control group consisted of 123 children without 77 antecedents of disease, born at 38-42 weeks of GA and with 2500-3500 g at birth. Are some of them born SGA?
Minor comments:
1. Table 1: Cesarean delivery à Cesarean delivery (%)
Mechanical ventilation à Mechanical ventilation (%) or (days) ?
Only one digit after the decimal place are required
Author Response
Dear editor and reviewer, we have added the following documents:
1) A new version of the manuscript.
2) A new version of the cover letter.
3) The English Editing Certificate.
4) The Authorship Change Form.
5) The original Ethical committee document
Please see the attachment
Response to Reviewer 1 Comments
Major comments
Point 1. The authors have to clarify whether this study is a “case-control” or “cohort” study. In the Abstract and Methods sections, it is described as case-control study. In the discussion section (P11 Line 315), it became “cohort”.
Answer 1: The case-control study is defined as an analytical observational study in which the subjects are selected on the basis of the presence of a disease or effect (cases) or not (controls) and, subsequently, the exposure of each of these groups is compared to one or more factors or characteristics of interest. We consider that the presence of EUGR (like prematurity or IUGR conditions) may represent a disease in itself, from the beginning, since it leads to poor growth in childhood, and may condition other comorbidities in the future. The EUGR is a status that conditions an early programming that continues during life. So, it may develop alterations in perinatal stage but also other associated pathologies at the prepubertal stage. For all of these reasons, this study has been considered as a case-control study as we have described in Material and Methods section. Moreover, to clarify this point, we have change the word “cohort” in the discussion, because it was referred to the group of children, not to a “cohort study”:
-Material and Methods (p2, line 81): "This is a descriptive, analytical, observational, case-control study".
-Discussion section: “First, it benefits from a large group of children from a same hospital” (p12, line 401); “comparison of data was performed from three well-defined groups of children” (p12, line 406); “this is a pioneer study of adipokines conducted with a group of premature children” (p12, line 409).
Point 2. The use of logistic regression analysis in this study is problematic. Logistic regression is a class of regression where the independent variable(s) is (are) used to predict the dependent variable. The dependent variable, therefore, occurred after the independent variable. The time sequence is reversed in the dependent and independent variables in this study. (ie. It is not a case-control study. The EUGR status occurred before the measurements of anthropometric parameters, blood pressure, metabolic markers and adipokines.)
Answer 2: As the reviewer indicates, normally, in the logistic regression, the outcome can be predicted from a number of predictor variables, but in this work, multiple (or multivariable) regression analysis did not search for a cause-effect, or the appearance of an event after an exhibition. Indeed, one of the best-known functions of the multivariate regression model is to assess the probability of the event in question (dependent variable) occurring as a function of certain variables that are presumed to be relevant or influential (independent variables). In these cases it is logical to assume that the dependent variable is later in time than the independent variables.
However, the multivariate model does not have a single predictive function, but also an explanatory function that allows us to know and quantify the degree of association using an OR adjusted for possible relevant or influential variants.
This relationship between a qualitative variable and continuous variables could have been made, with the same statistical value by means of an ANOVA adjusted for covariates. However, we decided to adjust it using a multivariate regression model to allow us a more simplified and understandable visualization of the results (table 4). The reason to do this was to find associations after the changes in individual variables at prepubertal age (anthropometric parameters, BP, metabolic biomarkers, or adipokines; independent variables) comparing with the antecedent of prematurity and/or EUGR (dependent variables) after adjustments.
To clarify the use of logistic regression analysis in our study, we have modified the following paragraphs:
-In Material and Methods section, Statistical Analysis (p4, lines 177-180): “The relations between EUGR and prematurity conditions with anthropometric parameters, BP, metabolic biomarkers and adipokines changes at prepubertal age were explored using a logistic regression model, estimating odds ratios (OR) values and 95% confidence intervals (95% CI).
-In Results section (p9, lines 253-255): “Table 4 displays the logistic regression analyses performed to identify the association between the anthropometry, BP and metabolic variables (including adipokines) of prepubertal children with the antecedents of prematurity and the EUGR condition”.
Point 3. In Table 3, correlation analysis should also be performed in the whole study population and then those in the subgroups.
Answer 3: As the reviewer indicates, we performed a correlation analysis in the whole study population and then in the subgroups of children. We have added in the Table 3 of the manuscript, the significant correlation coefficients of the global sample. We have included information about this point in the Results section (p8, lines 233-235): "After the correlation analysis in the whole study sample, correlations among plasma adipokines and anthropometric measurements, BP or metabolic parameters in different subgroups were made. Correlations that were significant are indicated in Table 3".
We have included the correlation between adiponectin and resistin in Table 3, so, the following paragraph has beeb added in the Discussion Section (p12, lines 388-396): "It must be also highlighted that a significant inverse correlation between adiponectin and resistin in EUGR children was found (Table 3). The interdependency between plasmatic adipokines’s profiles has been uncommon studied [65]. It has reported that the relation adiponectin/resistin index, may be an useful integrated determinant for diagnostic of insulin resistance or metabolic syndrome [66] and appears to play a role in inflammation pathways [13]. In fact, adiponectin exerts an anti-inflammatory protective effect, whereas resistin may have pro-inflammatory properties [67]. Thus, an inverse correlation between both adipokines leading to hypoadiponectinemia and hyperesistinemia may be expected in an "inflamed status" as could occur in prematurity and impaired growth".
Point 4. In Table 4, multiple logistic regression analysis is an accepted statistical method for assessing the association between an anticedant characteristic (risk factor) and a quantal outcome (probability of disease occurrence), statistically adjusting for potential confounding effects of other covariates. The EUGR status is not an outcome occurred after the risk factors listed in this table.
Answer 4: As table 4 represents the results after the multiple regression analysis, we have answered to this comment in point 2 to explain why this analysis was done.
Results about the relation between adiponectin and the condition of prematurity without extrauterine growth restriction were not reported in the previous manuscript because the p-value of this relation was > 0.25 in the simple logistic regression analysis. However, we have included this variable (adiponectin) in the multivariate regression analysis, and the results have been: 1 (0.98, 1.01) (Adjusted OR and 95% IC). We have added these data in Results Section, Table 4.
Point 5. In Table 1 and Table 2, there were differences in birth weight, prenatal steroid use, mechanical ventilation, age at prepubertal measurement and sex distribution between the EUGR and PREM children. The authors have to address whether these differences may result in the changes in prepubertal measurement in their discussion.
Answer 5: The authors agree with this comment and this could be a limitation related with all these antecedents. However, to avoid these effects, children recruited were assessed not only for the variables included in this study, but also for the neurologic, infections or respiratory evolution, and there were no clinical differences between them during these years of follow-up.
Regarding the differences in prenatal steroid use and mechanical ventilation between EUGR and PREM groups, we have added this clarification in Discusion Section (p10, lines 309-314): “The differences in prenatal steroid use and mechanical ventilation between EUGR and PREM children could result in changes in adipokines levels specially during the perinatal period and maybe at the prepubertal stage. For this reason we considered these variables and to follow if there were any sequelae in these children related. Notwithstanding, it has been reported that values of adiponectin, leptin or resitin are not affected by steroid treatment [38] or mechanical ventilation [39] in critical ill adult patients.”
In relation to the differences in birth weight and sex at prepubertal stage, we have added some sentences to clarify how we have tried to avoid this influence or how these differences have been also reported with similar results to ours:
- In Material and Methods Section (statistical analysis) (p 4, lines 169-170) : “All the results were adjusted for sex, birth weight and age at the prepubertal stage.”
-In Discussion section (p4, lines 302-305): “EUGR children were predominantly male and showed a lower absolute birth weight (without reaching the 10th percentile) than PREM children (Table 1). These data may be expected, since a lower birth weight and male sex have been independently related to the presence of EUGR in recent research [9].”
In relation to the differences in age at recruitment, we have detailed that the inclusion criteria related to the range of age for evaluation (before puberty) were similar among the three groups. We have indicated that all the participants in this study were at prepubertal age. Specifically, they were children at scholar age without pubertal signs (Tanner I) and without sexual hormonal changes of puberty and this state are more important to metabolic changes than age itself. We have highlighted this point in several sections:
- In Material and Methods Sections (p2, line 92) we have added: “As the EUGR group, PREM children were selected at prepubertal age”.
In Discussion section (p10, lines 300-308) we have clarified these details to emphasize that the prepubertal status in scholars is more important than age to evaluate these children: "Sex and age may be influencing factors in adipokine levels [36], and some recent research have also highlighted that normal physiological changes of puberty itself may have an impact on levels of some adipokines [37]. EUGR children were predominantly male and showed a lower absolute birth weight (without reaching the 10th percentile) than PREM children (Table 1). These data may be expected, since a lower birth weight and male sex have been independently related to the presence of EUGR in recent research [9]. We selected scholar children to avoid this puberty influence. In our study, the EUGR and control groups were older than the PREM group, but all the participants were at scholar age and without pubertal signs, according to the inclusion criteria. Moreover, all the results were adjusted for sex and age at the time of evaluation at prepubertal age.”
Point 6. The control group consisted of 123 children without 77 antecedents of disease, born at 38-42 weeks of GA and with 2500-3500 g at birth. Are some of them born SGA?
Answer 6: None of the control children were SGA. Both preterm or control children with the diagnosis of IUGR or SGA, defined as fetuses or newborns who had failed to achieve normal weight based on previous growth measurements in pregnancy, and with an estimated fetal weight less than the 10th percentile for gestational age (ACOG Practice Bulletin No. 204: Fetal Growth Restriction. Obstet Gynecol 2019) were excluded in this study. We have clarified this point in:
- Material and Methods (p2, lines 92-96): “The control group consisted of 123 prepubertal children meeting the following inclusion criteria: born at term with adequate birth weight for GA (2500–3500 g, 38–42 week’s GA) and born in the same period as children of the other two groups, without underlying conditions and any remarkable disease history”.
- Material and Methods (p3, lines 113-117): “Children with the diagnosis of intrauterine growth restriction (IUGR) or small gestational age (SGA), defined as fetuses or newborns who had failed to achieve normal weight based on previous growth measurements in the pregnancy, and with an estimated fetal weight that is less than the 10th percentile for gestational age [22]), were excluded in this study. Fetal growth was evaluated using anthropometric centiles for gestational ages 24 to 42 weeks from Yudkin et al. [23]”.
Minor comments:
Table 1: Cesarean delivery à Cesarean delivery (%); Mechanical ventilation à Mechanical ventilation (%) or (days)?
Answer: We have clarified these points in Table 1.
Cesarean delivery: Date is expressed as percentages of subjects (%)
Mechanical ventilation (%): Date is expressed as percentages of subjects (%)
Only one digit after the decimal place are required
Answer: Following the instructions of the reviewer, we have changed the number of digits after the decimal to one in all the tables.

Reviewer 2 Report
The study concerning plasma adipokines profile in prepubertal children with a history of prematurity or extrauterine growth restriction seems to be quite interesting, however many concerns are pointed below:
Abstract:
The Authors should present their results in a more precise and orderly manner considering the obtained values of the tested parameters and the p-value. Authors must conclude carefully and do not indicate the strong statements due to many limitations.
Introduction;
The introduction is too general and does not present the purpose of the research. Authors should briefly provide literature data on adipokine studies in children with a history of prematurity or EUGR with regard also to previous studies (reference 14). Why did the previous research of the Authors require continuation?
Materials and Methods:
- The inclusion criteria for EUGR are unclear - line: 48 (below P10), line: 70 (above P10)
- Authors did not specify the inclusion criteria such as BMI. Both studied groups must be described in details. In both groups examined, a number of children were overweight or obese. Normal body weight is usually when BMI Z-score <-1 +1> but in EUGR as well as PREM groups it can be seen that the BMI Z-score is definitely above 1, especially in the PREM group (-2.01, 3.52, interquartile range). It is known that abnormal adipokine profile may be observed in obese subjects (children, adults) (J Pediatr Gastroenterol Nutr. 2016 Jan;62(1):122-9; Horm Mol Biol Clin Investig. 2014 Apr;18(1):37-45), therefore it is important how many children in studied groups were overweight and obese? Authors should accurately describe the criteria for overweight and obesity.
-How was the control group recruited?
-In this section has no information on samples collection (blood, serum, plasma), kits (manufacturer, country, city), intra- and inter precision for assays, analytical equipment (manufacturer, country, city) etc. The number of repetitions for the assessments must be specified.
-Dietary intake in studied groups will be useful in assessment of relations between adipokines and anthropometric parameters as well as lipid profile. Especially, that these groups consisted of obese and non-obese children.
- There is a lack of accurate data (number) regarding the consent of the Ethical Committee.
Results:
- Exact p-values should be given in the description of the results, especially for adipokines.
- The number of decimal places for the values in the tables 1,2 should be corrected.
-Table 4 - change the order of the groups as in Tables 1-3,
- Authors should explain why the PREM group has no data on the relation between adiponectin and the condition of prematurity without extrauterine growth restriction (Table 4, not applicable)?
Discussion:
-The discussion should be without repetitions from the Results section (Table 1, Figure 1).
- line 236: The Authors write that “ Many different factors may influence adipokine values, such as age, gender, race, BMI and the degree of adiposity”. In Table 2, the studied groups differ significantly in terms of age, sex as well as BMI (obese and non-obese children). The authors should discuss adipokine profiles in the PREM and EUGR groups in the context of these differences and demonstrate that this could be a study limitation. The causes of obesity and the frequency of overweight and obesity in children with history of prematurity or EUGR should also be considered.
The Conclusions section: more clearly and reflecting the results obtained by the Authors. The same in the Abstract section.
English needs to be checked throughout the manuscript.
Abbreviations should be defined in the text (BP)
Author Response
Dear editor and reviewer, we have added the following documents:
1) A new version of the manuscript.
2) A new version of the cover letter.
3) The English Editing Certificate.
4) The Authorship Change Form.
5) The original Ethical committee document
Please see the attachments
Response to Reviewer 2 Comments
Point 1. Abstract. The Authors should present their results in a more precise and orderly manner considering the obtained values of the tested parameters and the p-value. Authors must conclude carefully and do not indicate the strong statements due to many limitations.
Answer: After your considerations, we have revised the order of the results and we have change the conclusion
Point 2. Introduction. The introduction is too general and does not present the purpose of the research. Authors should briefly provide literature data on adipokine studies in children with a history of prematurity or EUGR with regard also to previous studies (reference 14). Why did the previous research of the Authors require continuation?
Answer: Thank you for your comments. We have clarified the aim of this study at the end of the introduction and we have include other references related with adipokine studies in other samples with similar evolution or risk factors based in alterations on the adipose tissue during the perinatal stage. However, there is scarce literature about the role of the antecedent of EUGR in the later adipokines status, although it seems to be similar to children with obesity or IUGR history. In a previous research of this group (Ortiz et al. Nutrition 2014), children with a history of EUGR exhibited a distinctive adipokine profile with respect to healthy children born at term but we had the limitation that prematurity was not considered as a possible independent factor involved, so it was necessary to include another group of premature children without EUGR and IUGR condition and to compare results among the three groups.
We have included these comments in Introduction section (p2, lines 68-78): “However, there is scarce literature about the role of the antecedents of EUGR in the later adipokines status, although it seems to be similar to children with obesity or IUGR history. In a previous research of this group, children with a history of EUGR exhibited a distinctive adipokine profile with respect to healthy children born at term [19]. However, prematurity condition was not considered as a possible independent factor involved in the adipokines profile, as some studies have reported [20]. It is necessary to clarify whether changes in adipokine levels may be affected by prematurity itself. Therefore, the aim of this study was to evaluate serum concentrations of selected adipokines (adiponectin, resistin and leptin) in premature children with EUGR, those without EUGR and healthy children born at term; and to determine the associations between anthropometric and metabolic parameters and the conditions of EUGR and prematurity at the prepubertal stage”.
Point 3. Materials and Methods
- The inclusion criteria for EUGR are unclear - line: 48 (below P10), line: 70 (above P10)
Answer: As you indicated, we have revised the paragraphs for inclusion criteria. In our work, EUGR was defined as weight < P3 according to the age and sex at 36 weeks PMA and at discharge from the neonatal unit. We use the "true concept of EUGR" (Figueras et al. Eur J Nut 2020), which refers to cases of EUGR without evidence of fetal growth impairment (SGA or IUGR at birth) but with an evolution to decrease in growth after birth. For this reason, one of inclusion criteria (applied to all the children) was to be children born with a weight above P10 for gestational age (GA). EUGR, PREM and CONTROL children with the diagnosis of IUGR or SGA, defined as fetuses or newborns who had failed to achieve normal weight based on previous growth measurements in pregnancy, and with an estimated fetal weight less than the 10th percentile for GA (ACOG Practice Bulletin No. 204: Fetal Growth Restriction. Obstet Gynecol 2019, 133, 97-109), were excluded in this study. In order to assess fetal growth, centiles for gestational ages 24 to 42 weeks from Yudkin et al. (Early Hum Dev 1987, 15, 45-52) were used.
To clarify better these concepts in the manuscript, we have added this information:
-In Materials and Methods, Study participants (p2, lines 83-92): "The first was the EUGR group, with 38 children with the following criteria: prepubertal children born at ≤ 32 weeks of GA with a weight above P10 for GA at birth, and meeting the definition of "true-EUGR" based on the different concepts proposed in the literature (weight < P3 at 36 weeks-PMA and at discharge from the neonatal unit) [9]… The PREM group included a total of 50 children without puberty signs born at ≤ 32 weeks of pregnancy with a weight above P10 for GA at birth, and appropriate postnatal growth defined as weight ≥ P3 at 36 weeks-PMA and at discharge from the neonatal unit."
-In Materials and Methods, Clinical history and physical examination (p3, lines 113-117): "Children with the diagnosis of intrauterine growth restriction (IUGR) or small for gestational age (SGA), defined as fetuses or newborns who had failed to achieve normal weight based on previous growth measurements during pregnancy, and with an estimated fetal weight less than the 10th percentile for GA [22], were excluded in this study. Fetal growth was evaluated using anthropometric centiles for gestational ages 24 to 42 weeks from Yudkin et al. [23]."
- Authors did not specify the inclusion criteria such as BMI. Both studied groups must be described in details. In both groups examined, a number of children were overweight or obese. Normal body weight is usually when BMI Z-score <-1 +1> but in EUGR as well as PREM groups it can be seen that the BMI Z-score is definitely above 1, especially in the PREM group (-2.01, 3.52, interquartile range). It is known that abnormal adipokine profile may be observed in obese subjects (children, adults) (J Pediatr Gastroenterol Nutr. 2016 Jan;62(1):122-9; Horm Mol Biol Clin Investig. 2014 Apr;18(1):37-45), therefore it is important how many children in studied groups were overweight and obese? Authors should accurately describe the criteria for overweight and obesity.
Answer: We appreciate your comment. The BMI was not an inclusion criteria because growth evolution could be different in these children and could be related with their perinatal condition. Because of that, the inclusion criteria were related with measurements in perinatal stage to establish EUGR or prematurity criteria, but not with BMI that could be related by the adipose tissue implication during this evolution. So, to add this information, we have included the number of children with obesity in each group, as well as we have emphasized the lower growth.
To evaluate how many children in studied groups were overweight and obese, international classification by Cole et al. (BMJ 2000, 320, 1240-1243) was used. Obesity was defined as BMI centile above P97, specific for age and sex, and referred to the growth charts by Hernández et al. 1988. None of the children was obese in the EUGR and Control groups, and 5 children (10%) were obese in the PREM group. We have added these data in the manuscript (Table 2).
In the present study, EUGR children were no obese, and they exhibited more changes in adipokines levels despite lower anthropometric values and lower BMI z-score than those without EUGR or control children. In contrast, some children with a history only for prematurity were now obese. It seems that the role of the adipose tissue and the pathways to develop metabolic alterations may be different in these groups. In fact, associations between a fast childhood weight gain and higher adiposity and higher risk metabolic markers have been reported in preterm children and adolescents (Duncan et al. 2017, Embleton et al 2016).
To clarify this point, we have added the following paragraphs:
-In Material and Methods section, Clinical history and physical examination (p3, lines 132-134): "Obesity was defined as BMI > P97 at the prepubertal stage, according to Cole et al [18.1], specific for age and sex and referred to the growth charts by Hernández et al. 1988".
-In Results section, Table 2: “Obesity (%): EUGR group 0.0a; PREM group 5.0b; Control group 0.0a; P-value < 0.001”.“Obesity was defined as BMI > P97 for age and sex”.
- How was the control group recruited?
Answer: In methods section, we have added some details related to the recruitment of control group (p2, lines 88-94): “The control group consisted of 123 prepubertal children meeting the following inclusion criteria: born at term with adequate birth weight for GA (2,500–3,500 g, 38–42 week’s GA) and born in the same period as children of the other two groups, without underlying conditions and any remarkable disease history. These children were attended in the hospital’s Outpatients Unit at prepubertal age, on suspicion of mild illness that was later ruled out and requiring blood test. All the analyses showed that controls were free of any disease.”
- In this section has no information on samples collection (blood, serum, plasma), kits (manufacturer, country, city), intra- and inter precision for assays, analytical equipment (manufacturer, country, city) etc. The number of repetitions for the assessments must be specified.
Answer: As the reviewer requested, we have included specific information for this point. We did not include it before to prevent the manuscript from exceeding the word limit.
-In Materials and Methods Section, "Biochemical and adipokine analysis" (p4, lines 138-163): “Blood samples were collected in all the groups of children at 09.00 a.m. after a 12.00-hour overnight fast and at rest while lying down and using an indwelling venous line (median cubital vein) to draw a two 3-ml sample, which was used to extract serum and plasma, respectively. Serum samples were analyzed within 2 h of collection, while plasma samples were divided into aliquots and frozen at -80ºC until analysis."
Serum total cholesterol (TC) (CV≤3%), high-density lipoprotein cholesterol (HDL-C) (CV≤4%), low-density lipoprotein cholesterol (LDL-C), triglycerides (TG) (CV≤5%) and markers of carbohydrate metabolism, glucose (CV≤5%) and insulin(CV≤7%) were measured. IR was calculated by the homeostatic model assessment index (HOMA-IR=insulin (mU/l) x glucose (mmol/l)/22.5). Follicle stimulating hormone, luteinizing hormone, estradiol and testosterone were analyzed to confirm the prepubertal stage in all the children.
Analyses were performed by standardized laboratory methods and external and internal quality controls were performed according to hospital protocol. Autoanalyzers Architect c16000 and i2000SR (Abbott Diagnostics® Spain) were used for the measurements of general biomarkers (glucose, TC, HDL-C, LDL-C, TG) and general hormones (insulin), respectively.
Adiponectin, resistin and leptin plasma concentrations were measured using the cytometer Luminex® X MAP™ Technology (Labscan™ 100) in 96-well filter plates. It has a multi-analyte simultaneous detection system and uses LINCOplex assay kits. Each kit used has been validated for sensitivity, recovery, linearity, precision, and specificity. An immunoassay on the surface of polyethylene fluorescent-coded microspheres was performed. This surface is covered with a specific capture antibody that selectively binds to the adipokine, simultaneously introducing a marked detection antibody. This reaction mixture is incubated and, via a laser diode signal of each microsphere, is identified and quantified. All the analytes have been individually and in combination tested to ensure their compatibility with the multiplex system (Millipore technical protocols). The number of repetitions for the assessments of the general metabolic biomarkers and those of adipokines values were of one and two, respectively.
- Dietary intake in studied groups will be useful in assessment of relations between adipokines and anthropometric parameters as well as lipid profile. Especially, if these groups consisted of obese and non-obese children.
Answer: Standard food frequency intake and a 24-hour diet recall method questionnaires (designed in the Nutritional Institute of the University of Granada) were performed to collect information on food habits in the three groups at prepubertal age. A computer program estimated the daily energy and fibre intake and dietary macronutrient composition. The dietary information obtained was compared against the charts included in the Guide of Healthy Food Habits designed by the Spanish Society of Community Nutrition [Gil et al 2010].)
We found that daily caloric intake was not different between groups (EUGR group: 1855.8 Kcal/d ± 546.2 vs PREM group: 1755.5 ± 510,8 Kcal/d vs CONTROL group: 1585.5 ± 122.5 Kcal/d; p-value 0.354).
In relation with macronutrients, we considered that the results were not relevant because the error in intake estimation is usually high, especially when it is divided into subgroups. Moreover, there were little differences between groups that did not differ essentially between them comparing with international recommendations. Lipid intake was higher in the EUGR group than in the PREM group (EUGR group: 34.9 ± 6.2% (72 ± 12.7 gr/d) vs PREM group 22.3 ± 5.4% (46 ± 11.1 g/d); p <0.001), and similar to the CONTROL group (EUGR group: 34.9 ± 6.2% (72 ± 12.7 gr/d) vs Control group 33.9 ± 3.3% (70 ± 6.9 gr/d, p 0.785). Protein intake in the EUGR group was similar to that in the PREM group (EUGR group: 18.3 ± 3.2% (85 ± 14.8 g/d) vsPREM group: 19.3 ± 3.5% (89.7 ± 16.3 gr/d), p= 0.208), and higher than in the Control group (EUGR group: 18.3 ± 3.2% (85 ± 14.8 g/d) vs Control group: 13.5 ± 2.3% (62.8 ± 5 g/d), p < 0.001). The PREM group showed the highest carbohydrate intake (58.5 ± 4.8%, 271.1 ± 22.2 g/d, p < 0.001), followed by the EUGR group (46.8 ± 5.4%, 216.9 ± 25.3 g/d, p= 0.01) and the Control group (38.7 ± 2.9%, 179.5 ± 13.6 g/d).
We did not include these data in the manuscript because it would be too large, though it could be included in other article related with other metabolic variables directly related with intake. Although dietary consumption during childhood may be determinant in the growth and adult health (Wiedmeier et al. 2010), we consider that the changes in adipokines levels in premature children may be induced by the poor growth during the neonatal period, rather than by their dietary intake or values of BMI. In the present study, it does not seem that diet is related with the results in plasma adipokines, at least in EUGR group, in which there are no obese and there are not relevant results since there are not differences in energy and the lipid intake is the higher. They exhibited more changes in adipokines levels, but maintained lower anthropometric values and had lower BMI z-score than those without EUGR or control children. This premise is consistent with previous studies in non-obese children indicating there were no associations between dietary factors (such as intakes of total energy, fibre, fat, saturated fat, carbohydrates and proteins, or other selected nutrients) and leptin, adiponectin, or resistin concentrations (Al-Daghri et al. Endocr J 2012) (Aeberli et al. Int J Vitam Nutr Res 2009) (Hakanen et al Pediatrics 2004). Moreover, a recent study evaluating if the dietary pattern differs between healthy normal-weight children and, normal-weight children suffering from metabolic abnormalities (higher plasma leptin levels and higher leptin to adiponectin ratio), found no differences regarding the intake of total calories and macronutrients between the two groups (Nier et al. Nutrients 2019).
However, if the reviewer considered, we could add all these results in a complementary table or in the manuscript adding information in methods, results and discussion if the reviewer and editor consider, as we expose below:
-In Material and Methods, we will add: “2.4. Dietary assessment: “Standard food frequency intake and a 24-hour diet recall method questionnaires were realized to collect information on food habits in the three groups at prepubertal age. A computer program estimated the daily energy and fibre intake and dietary macronutrient composition. The dietary information obtained was compared against the charts included in the Guide of Healthy Food Habits designed by the Spanish Society of Community Nutrition [Gil Hernandez et al 20101].
-In Results section (after Table 2): “Regard to the nutritional assessment, we found that daily caloric intake was not different between groups (EUGR group: 1855.8 Kcal/d ± 546.2 vs PREM group: 1755.5 ± 510,8 Kcal/d vs CONTROL group: 1585.5 ± 122.5 Kcal/d; p-value 0.354). In relation with macronutrients, results were not relevant. There were little differences between groups that did not differ essentially comparing with international recommendations (results not shown)”.
- In Discussion section: “Many different factors may influence adipokine values, such as race, BMI and the degree of adiposity [34]. Although dietary consumption during childhood may be determinant in the growth and adult health [Wiedmeier et al. 20102], in the present study, diet or BMI seems unrelated with the results in plasma adipokines, at least in EUGR group in which there was no obese and there were not relevant nutritional results since there were not differences in energy intake. EUGR children exhibited more changes in adipokines levels but maintained lower anthropometric values, and had lower BMI z-score than those without EUGR or control children. So, the changes in adipokines levels in premature children may be induced by an adipose tissue dysfunction probably related to the poor growth during the neonatal period, rather than by their dietary intake or values of BMI. This premise is consistent with previous studies in non-obese children indicating there were no associations between dietary factors and adipokines [Aeberli et al. 20093]. Moreover, a recent study evaluating if the dietary pattern differs between healthy normal-weight children and, normal-weight children suffering from adipokines changes, found no differences regarding the intake of total calories and macronutrients between the two groups [Nier et al. 20194]. In contrast, an higher proportion of obesity was found in premature children without EUGR than in the other children. A fast catch-up growth later in these preterm children without EUGR, who exhibited a similar BMI z-score that those healthy born at term, could have lead an increased adiposity and an abnormal body composition. In fact, associations between a rapid childhood weight gain and higher adiposity and higher risk metabolic markers or alterations in adipokines values, have been reported in preterm children and adolescents [20,35].
- There is a lack of accurate data (number) regarding the consent of the Ethical Committee.
Answer: we have included the accurate data by the Ethical committee and we have added the original document in Spanish in complementary documents.
Material and Methods (p3, line 101-106): “This study was conducted in accordance with the Declaration of Helsinki and was approved by the Institutional Hospital Ethical Committee (protocol no 228, ref 2466; 2014). The selected subjects were incorporated into the study after all the inclusion criteria were fulfilled, and informed written consent from the parents or legal guardians of the children was obtained. Confidentiality of all personal information was protected. Access to medical data was obtained conforming to the hospital ethical standards”.
Point 4. Results:
- Exact p-values should be given in the description of the results, especially for adipokines.
Answer: we have added these data in the description of the results as the reviewer has suggested.
-In Result Section, Figure 1 (p8, lines 230-232): “Adiponectin: EUGR vs PREM P<0.001, EUGR vs CONTROL P<0.001; PREM vs CONTROL P= 0.004.
Leptin: EUGR vs PREM P=0.048, EUGR vs CONTROL P=0.214, PREM vs CONTROL P=0.029.
Resistin: EUGR vs PREM P= 0.007, EUGR vs CONTROL P<0.001, PREM vs CONTROL P<0.001"
- The number of decimal places for the values in the tables 1,2 should be corrected.
Answer: The number of decimal places has been corrected in the tables.
-Table 4 - change the order of the groups as in Tables 1-3,
Answer: Sorry, there´s was a mistake. We have reorganized the order of the groups in tables 1-3, for table 4.
- Authors should explain why the PREM group has no data on the relation between adiponectin and the condition of prematurity without extrauterine growth restriction (Table 4, not applicable)?
Answer: These data was not reported in the previous manuscript because the p-value of the relation between adiponectin and the condition of prematurity without extrauterine growth restriction was > 0.25 in the simple logistic regression analysis. However, we have included this variable (adiponectin) in the multivariate regression analysis, and the results have been: 1 (0.98, 1.01) (Adjusted OR and 95% IC). As the reviewer requested, we have added these data in Results Section, Table 4.
Point 5. Discussion:
-The discussion should be without repetitions from the Results section (Table 1, Figure 1).
Answer: We appreciate your comments, and have deleted some results from the discussion.
- line 236: The Authors write that “ Many different factors may influence adipokine values, such as age, gender, race, BMI and the degree of adiposity”. In Table 2, the studied groups differ significantly in terms of age, sex as well as BMI (obese and non-obese children). The authors should discuss adipokine profiles in the PREM and EUGR groups in the context of these differences and demonstrate that this could be a study limitation. The causes of obesity and the frequency of overweight and obesity in children with history of prematurity or EUGR should also be considered.
Answer: As the reviewer has indicated, we have already commented them extensively.
Point 6. The Conclusions section: more clearly and reflecting the results obtained by the Authors. The same in the Abstract section.
Answer: Thank you for your suggestion. We have adapted the conclusion as similar as in the abstract section.
Point 7. English needs to be checked throughout the manuscript.
Answer 7: The manuscript was edited for English language usage, grammar, spelling and punctuation by native English-speaking editors at Nature Research Editing Service, a service from Springer Nature, one of the world's leading research, educational and professional publishers that provides high-quality editing. We attached the certificate in complementary documents. Now, specialists in English editing have reviewed the new version of the manuscript again.
Point 8. Abbreviations should be defined in the text (BP)
Answer: Sorry, it was an error because SBP and DBP had been defined before. We have corrected it and revised all of the abbreviations.
References
1Gil Hernández A. Chapter 2. Ingestas dietéticas de referencia, objetivos nutricionales y guías; 1–65. Leis R. Chapter 9, Nutrición del niño de 1–3 años, preescolar y escolar; 227–56. In Tratado de nutrición. Tomo III. Nutrición humana en el estado de salud, 2nd ed.; Editor A. Gil Hernández. Editorial Panamericana; Publisher: Madrid, Spain, 2010.
2Wiedmeier J.E.; Joss-Moore L.A.; Lane R.H.; Neu J. Early postnatal nutrition and programming of the preterm neonate. Nutr Rev 2010;69:76–82
3Aeberli I.; Spinas G.A.; Lehmann R.; l'Allemand D.; Molinari L.; Zimmermann M.B. Diet Determines Features of the Metabolic Syndrome in 6- To 14-year-old Children. Int J Vitam Nutr Res 2009, 79, 14-23. DOI: 10.1024/0300-9831.79.1.14
4Nier A.; Brandt A.; Baumann A.; Conzelmann I.B.; Özel Y.;, Bergheim I. Metabolic Abnormalities in Normal Weight Children Are Associated with Increased Visceral Fat Accumulation, Elevated Plasma Endotoxin Levels and a Higher Monosaccharide Intake. Nutrients 2019, 18;11. pii: E652. DOI: 10.3390/nu11030652

Round 2
Reviewer 1 Report
This is a cohort study and the authors have to modify their study design.
Author Response
Dear editor and reviewers, we have added the following documents:
1) A third version of the manuscript.
2) A new version of the cover letter.
3) A document with information complementary about the sub-analysis without obese children.
We have used the red colour in these documents, so that new changes are easily visible to the editors and reviewers.
Please see the attachment
Point 1. This is a cohort study and the authors have to modify their study design.
Answer 1: As the reviewer indicates, we have modified the design of the study, and we have changed the words “cohort study” in Introduction and Material and Methods instead the words “case-control study”.
-Abstract (line 24): “The aim of this cohort study was…”
-Material and Methods (line 78): "This is a descriptive, analytical, observational, cohort study".
Additional information for editor and reviewers: María José de la Torre-Aguilar has been included in the authorship because she carried out the dietary intake work and she has participated in the new statistical study.

Reviewer 2 Report
The authors have included a number of corrections in the article, but there are still a few points to explain:
- Abstract: The Authors should present their results in a more precise and orderly manner considering the obtained values of the tested parameters and the p-value.
- Materials and Methods: Lincoplex kits for adipokines (manufacturer, country, city), intra- and inter precision for assays,
- Lines: 352-354 “In our study, the PREM group exhibited increased leptin levels (Fig.1) as well as higher values of insulin and HOMA-IR (Table 2), compared with the control group, suggesting fat tissue remodeling in preterm children even with normal BMI z scores.”
This is not entirely true because the group also includes several obese patients.
Since a relationship between obesity and the adipokine profile cannot be excluded in overweight patients, the Authors should analyze the results obtained in the PREM group excluding 5 obese patients. If the tendencies regarding these adipokines obtained in a group of 50 patients were confirmed, the Authors should take into account the statement that the participation of the obese patients in PREM group did not significantly affect the results obtained in this article. Changes in the adipokine profile are associated with prematurity.
- Line 395: I suggest rather “pro-inflammatory status” instead ”inflamed status”
- Lines 303,337,339,353,365,387,389 : Avoid in the Discussion section (Table 1) (Figure 1) etc.
- It is advisable to supplement the article with dietary data in Materials and Methods, Results (Table) and in the Discussion section. What computer program was used to evaluate the diet?
Author Response
Point 1. Abstract. The Authors should present their results in a more precise and orderly manner considering the obtained values of the tested parameters and the p-value.
Answer: After your considerations, we have included the most relevant data and reordered the abstract to get 200 words. Correlations have been eliminated from the abstract.
"Adipose tissue programming could be developed in very preterm infants with extrauterine growth restriction (EUGR), with an adverse impact on long-term metabolic status, as has been studied in intrauterine growth restriction patterns. The aim of this cohort study was to evaluate the difference in levels of plasma adipokines in children with a history of EUGR. A total of 211 scholar prepubertal children were examined: 38 with a history of prematurity and EUGR (EUGR), 50 with a history of prematurity with adequate growth (PREM), and 123 healthy children born at term. Anthropometric parameters, blood pressure, metabolic markers and adipokines (adiponectin, resistin, leptin) were measured. Children with a history of EUGR showed lower values of adiponectin (μg/mL) compared with the other two groups: (EUGR: 10.6 vs PREM: 17.7,P<0.001; vs CONTROL: 25.7,P=0.004) and higher levels of resistin (ng/mL) (EUGR: 19.2 vs PREM: 16.3, P=0.007; vs CONTROL: 7.1, P<0.001. The PREM group showed the highest values of leptin (ng/mL), compared with the others: PREM: 4.9 vs EUGR: 2.1,P=0.048; vsCONTROL: 3.2,P=0.029). In conclusion, EUGR in premature children could lead a distinctive adipokines profile, likely associated with an early programming of the adipose tissue, and increase the risk of adverse health outcomes later in life."
Point 2. Materials and Methods: Lincoplex kits for adipokines (manufacturer, country, city), intra- and inter precision for assays,
Answer: We have added the requested information rewriting the paragraph (lines 150-154): "To determine adiponectin (CV, 7.9%), resistin (CV, 6.0%), and leptin (CV, 7.9%) (Cat. HADK2–61K-B), LINCOplex kits of human monoclonal antibodies (Linco Research, St Charles, MO) were used on a Luminex 200 System (Luminex® X MAP™ Technology (Labscan™ 100), Luminex Corporation, Austin, TX). This is multi-analyte simultaneous detection system with a cytometer with 96-well filter plates."
Point 3. Lines: 352-354 “In our study, the PREM group exhibited increased leptin levels (Fig.1) as well as higher values of insulin and HOMA-IR (Table 2), compared with the control group, suggesting fat tissue remodelling in preterm children even with normal BMI z scores.”
This is not entirely true because the group also includes several obese patients.
Answer: To clarify it and according to the last statistic study of the parameters and adipokines profile excluding obese children, we have added the following phrase in the Statistical analysis (lines 178-179): “Moreover, a sub-analysis of the anthropometric parameters, BP levels, metabolic markers and adipokines profile excluding obese patients in all the groups were performed.”
After to consider these few obese children in Prem group, and in accordance with the results, we have modified the phrase avoiding the last comment to which the reviewer refers (Discussion section, lines 388-395): “In the present study, the PREM group exhibited increased leptin levels as well as higher values of insulin and HOMA-IR, compared with the control group. However, these differences were not found when the 5 obese children of the PREM group were excluded from the analysis, suggesting a higher risk of fat tissue remodelling probably related with increased BMI z-score in this group of children. In fact, the prevalence of the obesity has been associated with prematurity, and it has been reported that preterm children with higher adiposity and metabolic abnormalities, may have disproportionately higher values of fat mass and leptin [4, Sipola et al., Am J Epidemiol 20155]."
Point 4. Since a relationship between obesity and the adipokine profile cannot be excluded in overweight patients, the Authors should analyze the results obtained in the PREM group excluding 5 obese patients. If the tendencies regarding these adipokines obtained in a group of 50 patients were confirmed, the Authors should take into account the statement that the participation of the obese patients in PREM group did not significantly affect the results obtained in this article. Changes in the adipokine profile are associated with prematurity.
Answer: As the reviewer request, we have realized a new statistical study excluding children with obesity. There are some significant changes in values of insulin, HOMA-IR, and leptin levels in the PREM group (in which there was 5 obese), but the rest of parameters and p-values, are similar. We attach the relevant data after this statistical study in another complementary document to be reviewed in detail.
It is appropriate don´t ignore the prevalence of obesity found in the PREM group compared to the other two groups. In fact, the association between prematurity and obesity has been reported in some previous studies (Kopec et al., Diabetes Metab Syndr Obes 2017; Sipola-Leppänen et al., Am J Epidemiol 2015). For this reason, we have maintained the initial results in the manuscript. Obesity could be a consequence and it has not been considered an exclusion criteria; so we have added this new data after the sub-analysis performed without obese children in Material and Methods section (lines 178-179) as commented before, and the new values of leptin in PREM group after this sub-analysis in Results (lines 241-248), as well as the possible influence of BMI-Z score and obesity in metabolic and leptin changes of the PREM group in Discussion section (lines 388-395) as indicated.
Point 5: Line 395: I suggest rather “pro-inflammatory status” instead ”inflamed status”
Answer: As the reviewer suggests, we have changed the words “pro-inflammatory status” instead “inflamed status”.
Point 6: Lines 303,337,339,353,365,387,389: Avoid in the Discussion section (Table 1) (Figure 1) etc.
Answer: We have deleted the words “Table” or “Figure” in the Discussion section, as the reviewer indicates.
Point 7: It is advisable to supplement the article with dietary data in Materials and Methods, Results (Table) and in the Discussion section. What computer program was used to evaluate the diet?
Answer: As the reviewer indicates, we have added the information about dietary assessment in Material and Methods section. Moreover, a new table (Table 3) with dietary intake data has been added in Results section. This new information has been discussed in Discussion section.
The computer program used to evaluate the diet was “ODIMET®, Organizador dietético metabólico”, designed by the Pediatrics Department, Diagnosis and Treatment Unit for Metabolic Diseases, Santiago de Compostela University Clinical Hospital, Spain, 2008. This program is an online tool designed to calculate diets from more than 2000 products and foods with the maximum and validated information on their composition. We have clarified this point in the Material and Methods section.
-Material and Methods (lines 163-170): “2.4. Dietary assessment: “Standard food frequency intake and a 24-hour diet recall method questionnaires were made to collect information on food habits in the three groups at prepubertal age. The computer program “ODIMET® (Organizador Dietético Metabólico)”, designed by the Santiago de Compostela University Clinical Hospital (Spain 2008), was used to estimate the daily energy and fibre intake and dietary macronutrient composition. The dietary information obtained was compared against the charts included in the Guide of Healthy Food Habits designed by the Spanish Society of Community Nutrition [Gil Hernandez et al 20171].
-Results section (lines 222-236): “Regarding nutritional assessment, the Dietary Reference Intakes (DRIs) and the comparison by groups in macronutrients intake are shown in Table 3. Daily caloric intake was not different between groups. The PREM group showed the highest carbohydrate intake (p<0.001), followed by the EUGR group (p=0.010) and then the Control group (p<0.001). Lipid intake was higher in the EUGR group than in the PREM group (p<0.001), and similar to the CONTROL group (p= 0.785). Protein intake in the EUGR group was similar to that in the PREM group (p= 0.208), and higher than in the Control group (p<0.001)".
-Discussion section (lines 315-328): “Many different factors may influence adipokine values, such as race, BMI and the degree of adiposity [34]. Although dietary consumption during childhood may be determinant in the growth and adult health [Wiedmeier et al., Nutr Rev 20102], in the present study, diet or BMI seems unrelated with the results in plasma adipokines, at least in EUGR group, in which there was no obese and there were not relevant nutritional results since there were not differences in energy intake. EUGR children exhibited more changes in adipokines levels but maintained lower anthropometric values, and had lower BMI z-score than those without EUGR or control children. So, the changes in adipokines levels in premature children may be induced by an adipose tissue dysfunction probably related to the poor growth during the neonatal period, rather than by their dietary intake or values of BMI. This premise is consistent with previous studies in non-obese children indicating there were no associations between dietary factors and adipokines [Aeberli et al., Int J Vitam Nutr Res 20093]. Moreover, a recent study evaluating if the dietary pattern differs between healthy normal-weight children and, normal-weight children suffering from adipokines changes, found no differences regarding the intake of total calories and macronutrients between the two groups [Nier et al., Nutrients 20194].
Additional information for editor and reviewers: María José de la Torre-Aguilar has been included in the authorship because she carried out the dietary intake work and she has participated in the new statistical study.
New references included:
1Gil Hernández A. Tomo III. Nutrición humana en el estado de salud. En: Gil A (Ed), Tratado de Nutrición. 3ª edición. Panamericana, Madrid, ISBN: 978-84-9110-195-6; 2017.
2Wiedmeier J.E.; Joss-Moore L.A.; Lane R.H.; Neu J. Early postnatal nutrition and programming of the preterm neonate. Nutr Rev 2010;69:76–82
3Aeberli I.; Spinas G.A.; Lehmann R.; l'Allemand D.; Molinari L.; Zimmermann M.B. Diet Determines Features of the Metabolic Syndrome in 6- To 14-year-old Children. Int J Vitam Nutr Res 2009, 79, 14-23. DOI: 10.1024/0300-9831.79.1.1
4Nier A.; Brandt A.; Baumann A.; Conzelmann I.B.; Özel Y.;, Bergheim I. Metabolic Abnormalities in Normal Weight Children Are Associated with Increased Visceral Fat Accumulation, Elevated Plasma Endotoxin Levels and a Higher Monosaccharide Intake. Nutrients 2019, 18,11. DOI: 10.3390/nu11030652
5Sipola-Leppänen M.; Vääräsmäki M.; Tikanmäki M.; Matinolli H.M.; Miettola S.; Hovi P.; Wehkalampi K.; Ruokonen A.; Sundvall J.; Pouta A.; Eriksson J.G.; Järvelin M.R.; Kajantie E. Cardiometabolic risk factors in young adults who were born preterm. Am J Epidemiol 2015, 181, 861-873. DOI: 10.1093/aje/kwu443.
